# Discovery of the Relationship between Distribution and Aflatoxin Production Capacity of *Aspergillus*
*species* and Soil Types in Peanut Planting Areas

**DOI:** 10.3390/toxins14070425

**Published:** 2022-06-22

**Authors:** Shujuan Zhang, Xue Wang, Dun Wang, Qianmei Chu, Qian Zhang, Xiaofeng Yue, Mengjie Zhu, Jing Dong, Li Li, Xiangguo Jiang, Qing Yang, Qi Zhang

**Affiliations:** 1Xiangyang Academy of Agricultural Sciences, Xiangyang 441057, China; zhangshujuan1990@126.com (S.Z.); maysxuer@126.com (X.W.); chuqm_nky@163.com (Q.C.); lly1262094720@163.com (Q.Z.); zmj2280880980@163.com (M.Z.); xynkydongjing@163.com (J.D.); trustmeplease@126.com (L.L.); xfjxg1972@163.com (X.J.); qingyunsanhao@126.com (Q.Y.); 2Key Laboratory of Detection for Biotoxins and Key Laboratory of Biology and Genetic Improvement of Oil Crops, Ministry of Agriculture and Rural Affairs, Oil Crops Research Institute, Chinese Academy of Agricultural Sciences, Wuhan 430062, China; xiaofengl19870207@163.com; 3Zhejiang Mariculture Research Institution, Wenzhou 325000, China; 4Hubei Hongshan Laboratory, Wuhan 430061, China

**Keywords:** *Aspergillus species*, distribution, aflatoxin-producing capacity, soil types

## Abstract

In order to study the relationship between the distribution and aflatoxin production capacity of *Aspergillus species* and soil types, 35 soil samples were collected from the main peanut planting areas in Xiangyang, which has 19.7 thousand square kilometers and is located in a special area with different soil types. The soil types of peanut planting areas in Xiangyang are mainly sandy loam and clay loam, and most of the soil is acidic, providing unique nature conditions for this study. The results showed that the *Aspergillus sp.* population in clay loam (9050 cfu/g) was significantly larger than that in sandy loam (3080 cfu/g). The percentage of atoxigenic *Aspergillus* strains isolated from sandy loam samples was higher than that from clay loam samples, reaching 58.5%. Meanwhile the proportion of high toxin-producing strains from clay loam (39.7%) was much higher than that from sandy loam (7.3%). Under suitable culture conditions, the average aflatoxin production capacity of *Aspergillus* isolates from clay loam samples (236.97 μg/L) was higher than that of strains from sandy loam samples (80.01 μg/L). The results inferred that under the same regional climate conditions, the density and aflatoxin production capacity of *Aspergillus sp.* in clay loam soil were significantly higher than that in sandy loam soil. Therefore, peanuts from these planting areas are at a relatively higher risk of contamination by *Aspergillus sp.* and aflatoxins.

## 1. Introduction

*Aspergillus flavus* is one of the most important fungi causing global grain contamination [1]. Sampling investigations showed that *Aspergillus flavus* and aflatoxins were detected in stored corn and peanuts in many places in China [2]. Aflatoxin is a secondary metabolite produced by *Aspergillus* fungi, such as *Aspergillus flavus, Aspergillus parasiticus* and *Aspergillus nomius* [3]. Aflatoxin is highly toxic, mutagenic and carcinogenic, and long-term exposure to aflatoxin can induce primary liver cancer. In recent years, some studies have shown that aflatoxins can also cause cancer in the pancreas, kidneys, bladder and other organs, and may also lead to poor nutritional metabolism, immunosuppression and other pathologies [4,5,6]. Aflatoxin production capacity differs in different strains. According to various studies, more than 20 types of different aflatoxins have been found, and the main types of naturally occurring aflatoxins are AFB_1_, AFB_2_, AFG_1_, AFG_2_, etc. [7]. Aflatoxin B_1_ is recognized as the most toxic and carcinogenic natural mycotoxin, which was classified as a Class I carcinogen by the International Agency for Research on Cancer (IARC) of the World Health Organization (WHO) in 1993 [8].

Xiangyang is located in the middle and lower reaches of the Yangtze River. It now has formed large-scale peanut-producing areas such as Zaoyang, Xiangzhou, Yicheng and Gucheng, with a total planting area of nearly 66,667 hectares and a yield of 4400 kg per hectare, approximately. As the main peanut-producing area and distributing center in the Yangtze River Basin, Xiangyang has been included in aflatoxin contamination investigation and research many times [9,10,11]. According to these investigations, the level of *Aspergillus flavus* infestation and aflatoxin contamination in Xiangyang are slightly lower than the national average [11]. However, most of these studies only focus on the infestation of peanuts by *Aspergillus flavus* and less on the relationship between soil types and *A. flavus* distribution and the aflatoxin level. Since soil is directly in contact with plant roots and there is an exchange of nutrients, it has a great impact on the occurrence of aflatoxin contamination in crops [12]. Previous studies have pointed out that soil is the main source of aflatoxin contamination and *Aspergillus flavus* infection of peanuts and other crops [13,14,15,16,17,18], and the distribution, toxin production capacity of *Aspergillus flavus* and the degree of infestation of peanut vary greatly in different soil types [8,19]. Therefore, a study on the relationship among the distribution, toxin production capacity and contamination of *Aspergillus flavus* in the soil of peanut planting areas in Xiangyang and the local soil type is of guiding significance for the future agronomic management of fertilization, irrigation and *Aspergillus flavus* control in the peanut planting process.

Xiangyang has a unique geographical location and soil composition, which provides natural conditions for conducting research related to different soil types under the same climatic conditions. In this investigation, soil samples were collected from the peanut planting areas in Xiangyang, and the distribution and toxin production capacity of *Aspergillus species* in two types of soils were studied. The results illustrated the specificity of soil in Xiangyang and its connection with the distribution and aflatoxin production capacity of *Aspergillus sp.* and can be meaningful in providing more solutions for future development of prevention and control of aflatoxin contamination in peanuts.

## 2. Results

### 2.1. Isolation and Verification of Aspergillus Strains from Two Types of Soil

By the method described in Section 5.2.1, 116 *Aspergillus* strains which produced yellow-green spores were initially screened. The 116 strains obtained from the initial screening were cultured on DG-18 plates to obtain a single colony and then sent to a third-party testing institution (Beijing Prime Sequencing Company (Wuhan, China)) for molecular biological identification by sequencing with ITS universal primers. The sequence files were aligned in NCBI using the Blast program, and 99 isolates were identified with 99–100% similarity to the *A. flavus* strains in the database. The comparison results indicated that the isolates were dominated by *A. flavus*, with a possibility of small amount of *A. parasiticus* present.

### 2.2. Relationship between Soil Types and Distribution of Aspergillus Isolates

Based on soil properties, the soil samples collected from the four main peanut planting areas in Xiangyang could be classified into two categories which were sandy loam and clay loam. As shown in Figure 1, there were 19 sandy loam samples mainly distributed in the planting areas in Yicheng and Gucheng, while the clay loam samples were mainly in Zaoyang. The clay loam and the sandy loam were both found in Xiangzhou, with sample numbers of 7 and 7, respectively. The number of *Aspergillus sp.* colonies in sandy loam ranged from 0 to 12,000 cfu/g, and the average number was 3080 cfu/g (Table 1). Meanwhile the number of *Aspergillus sp.* colonies in clay loam was in the range of 0–24,000 cfu/g, and the average number was 9050 cfu/g (Table 1). It can be seen that the density of *Aspergillus sp.* in sandy loam was significantly lower than that in clay loam.

### 2.3. Aflatoxin-Producing Types in Different Soil Types

As shown in Figure 2, there were some differences in the distribution of toxigenic types of *Aspergillus isolates* between sandy loam and clay loam. On the whole, the strains producing AFB_1_, AFB_2_ and AFG_2_ accounted for the highest proportion in both types of soils, with 46.2% and 58.7%, respectively (Figure 2). These were followed by strains producing AFB_1_ in sandy loam and strains producing AFB_2_ in clay loam, with 23.1% and 28.3%, respectively (Figure 2). The percentage of *Aspergillus sp.* isolates only producing AFB_1_ in sandy loam was higher (19.2%) than that in clay loam soils (6.5%). The isolates producing both AFB_2_ and AFG_2_ were only found in sandy loam, while isolates producing both AFB_1_, AFB_2_ and AFG_1_ and *A. flavus* producing all four aflatoxins were only found in clay loam.

### 2.4. Aflatoxin Production Capacity of Aspergillus Isolates in Different Types of Soils

*Aspergillus sp.* isolated from the soil samples were cultured by the method described in Section 5.2.2, and then the content of aflatoxin in the culture solution was measured by LC. According to the toxin-producing level, *Aspergillus isolates* were classified into non-aflatoxin detected, low toxin and high toxin-producing strains, corresponding to the content of aflatoxin (AFT) of non-detected (N.D.), 0–100 μg/L and over 100 μg/L. The data in Figure 3 showed that the proportion of non-aflatoxin detected strains was higher in sandy loam samples than in clay loam samples. The proportions of low toxin-producing strains in clay loam (39.7%) and sandy loam (34.1%) were similar, while the differences in proportions of high toxin-producing strains in two types of soils were huge, with 39.7% in clay loam and 7.3% in sandy loam, respectively. It indicated the predominance of high toxin-producing strains in clay loam samples.

Analysis of the data in Figure 4 and Table 2 showed that more toxigenic strains were isolated from clay loam samples, and that the majority of strains isolated from sandy loam samples had a theoretical toxin production capacity in the range of 0 to 50 μg/L, with only two higher points exceeding 1400 μg/L. However, under theoretical conditions, most of the toxin-producing isolates from clay loam had more toxin-producing ability than the isolates from the sandy loam did. By analyzing the toxin production of the *Aspergillus sp.* isolated from the soil samples, it was found that the average toxin production of the *Aspergillus sp.* isolated from the clay loam samples (236.7 μg/L) was significantly higher than that of the isolates from the sandy loam samples (80.01 μg/L). According to the content of aflatoxin in the culture broth of isolates, the toxin-producing potential in each gram of soil sample was calculated. The results showed that one gram of sandy loam sample could theoretically produce 246.44 mg/L of aflatoxin under suitable conditions, and clay loam could produce 2144.58 mg/L of aflatoxin. Therefore, not only the density of *Aspergillus sp.* colonies in clay loam, but also the average production of aflatoxin by the strains was higher.

### 2.5. The Relationship between Soil Properties and the Number of Aspergillus sp. Colonies and Toxin Production Capacity

All soils samples were sent to a third-party testing institution (Wuhan Ziyu Testing Technology Co., Ltd. (Wuhan, China)) for detecting the major properties of soil. The results presented in Table 3 illustrated that there was no significant difference between sandy loam samples and clay loam samples in terms of the contents of total nitrogen, phosphorous and potassium. In addition, the R values presented in Figure 5 showed that the correlations between the contents of these three substances and *Aspergillus sp.* population were very low. Thus, the correlations of these properties and the propagation of strains were not attempted.

While the level of organic matter showed an extremely significant difference (*p* = 0.001) between sandy loam and clay loam samples, which might be related to the difference between *Aspergillus sp.* populations in two soil types. It can be seen from Figure 6 that the sandy loam samples had less organic matter than the clay loam samples, with 5.45% and 8.31%, respectively. Likewise, the mean propagule density of *Aspergillus sp.* in sandy loam samples was 3080 cfu/g, much lower than that in clay loam samples.

By measuring the pH value of the collected soil samples, there were 28 samples with pH < 7 (weakly acidic) and 7 samples with pH > 7 (weakly alkaline) out of 35 soil samples. It can be seen that the soils of the main peanut planting areas in Xiangyang were predominantly weakly acidic, and most of the soil samples had pH values in the range of 4.0–6.0 (Figure 7). The pH values of the clay loam samples were all less than 7 (4 to 6.5), which were all weakly acidic, while the pH values of the sandy loam samples ranged from 4.0 to 8.2, of which 36.8% was weakly alkaline, 63.2% was weakly acidic. The mean pH value of sandy loam samples (pH = 6.14) was slightly higher than that of clay loam samples (pH = 5.33). As can be seen from Table 4, the average number of *Aspergillus sp.* colonies from the weakly acidic soil samples (7100 cfu/g) was significantly higher than that from the weakly alkaline soil samples (1086 cfu/g). The range of aflatoxin produced by the strains isolated from weakly acid soil samples was much wider than that from weakly alkaline soil samples, and the average amount of aflatoxin produced by strains from weakly acid soils was also much higher than that from weakly alkaline soils.

To clearly show the correlations between the toxin-producing capacity and soil pH, the toxin-producing capacity of *Aspergillus* isolates was exhibited in two groups (Figure 8). Both in low and high toxin-producing groups, the toxin-producing capacity of the strains isolated from soil samples with low pH was higher than that from soil samples with high pH, regardless of the soil types from which the strains were isolated.

## 3. Discussion

### 3.1. Characteristics of Aspergillus Isolates Differ in Two Typical Soil Types

The soil in the major peanut planting areas in Xiangyang was mainly divided into sandy loam and clay loam. The sandy loam samples came from areas located along the Han River and the Tangbai River, while the clay loam soil came from the hilly or hill areas far away from rivers. According to Zhang Chushu’s survey of aflatoxin-producing *Aspergillus sp.* from peanut field soils in four agroecological zones of China, there are nearly 94.2% of strains identified as *A. flavus* and 5.8% identified as *A. parasiticus* in main peanuts planting areas [20], which is very similar to the sequencing results in this survey. The density of *Aspergillus sp.* in clay loam was higher than that in sandy loam, which is consistent with in the results presented in Zhang Xing’s [11] study conducted in China. Plus, Jaime-Garcia, R. and Cotty believe that soil types with large clay and small sand contents provide favorable conditions for large aflatoxin contamination in southern Texas soils, where the amount of *A. flavus* populations was positively correlated with clay content [21], probably because clay loam has good water retention properties, which is most conducive to the growth of *A. flavus*. In this study, the texture and particle size of the clay loam in peanut production areas of Xiangyang were more biased towards clay, with better water retention performance. In addition, peanuts are usually planted and grown in the summer in Xiangyang, thus, the high temperature and the good water retention of the soil work together, prompting the proliferation of *A. flavus* in clay loam.

According to the test results of the toxin-producing culture of the isolates, the dominant toxin-producing types of isolates from sandy loam and clay loam samples were similar, but the types in clay loam samples were more abundant than those that in sandy loam (Figure 3), indicating that there were also some differences in the types of *Aspergillus sp.* in the two soil types. In terms of aflatoxin production capacity, the distribution of the atoxigenic *Aspergillus strains* and the toxin-producing strains in the two types of soil was significantly different. Among the strains from clay loam samples, high toxin-producing strains with toxin production over 100 μg/L accounted for 39.7%, which was much higher than that in sandy loam samples. Therefore, the clay loam was liable to produce more aflatoxins under the high temperature conditions during the growth period of summer peanuts, implying that the peanuts were more likely to be contaminated by aflatoxins. As to the differences in the distribution of the *Aspergillus* strains with different toxin production in clay loam and sandy loam, the reason may be that the clay loam in peanut planting areas of Xiangyang was more suitable for the growth of high toxin-producing strains, or the presence of a large number of non-aflatoxin detected strains in sandy loam may have formed a competitive relationship with high toxin-producing strains and inhibited the growth of high toxin-producing strains. Zhang Chushu [10] and Wu Linxia [19] both proposed the competitive inhibition of non-aflatoxin detected strains against the reproduction of high toxin-producing strains.

The average toxin production of *A. flavus* strains isolated in clay loam was about three times higher than that of sandy loam (Table 2). The reasons, on the one hand, may be the high percentage of toxin-producing strains in the composition of microflora in clay loam under the unique natural conditions of Xiangyang. On the other hand, probably because of the good water retention of clay loam, *A. flavus* has a stronger ability to produce toxins under high temperature and water activity. The present study showed that the soil in peanuts planting areas in Xiangyang was mainly loam, and the data from the study conducted by Wu Linxia [19] indicated that the loam soil accounted for a relatively high proportion of soil samples with excess aflatoxin, reaching 61.94%. In Figure 5, there are two abnormally high points in the toxin production of *A. flavus* strains isolated from sandy loam samples, both of which produced more than 1000 μg/L aflatoxin. These two strains came from the soil samples from Yicheng, and aflatoxin production capacity of these two strains differed from other strains isolated from this sampling site. It might because the purchased peanut seeds carry high toxin-producing strains and which spread to soil during planting. Another hypothesis is that high toxin-producing strains are originally existed in the natural microbial environment of this sampling site, but only accounting for a very small number due to a significant advantage in the proportion of non-aflatoxin detected strains, which inhibited the growth of high toxin-producing strains.

### 3.2. The Properties of Soil May Affect the Number of Colonies and the Formation of Strain Group Structure

It has been reported that carbon, nitrogen, pH and other environmental factors could impact *A. flavus* density and aflatoxin production [22,23]. In this study, some main properties of soil were detected as well, and the results showed that organic matter and pH may have influenced the density of *Aspergillus sp.* and the strain group structure.

By assessing the organic matter content, the *Aspergillus* strains population may be associated with the content of organic matter in soil. The content of organic matter in clay loam was higher than that in sandy loam samples, which may have led to the higher number of colonies in clay loam samples. Large content of organic matter can provide more nutrition to the proliferation of *Aspergillus* strains. A similar conclusion was given in a study conducted by Zablotowicz et al. [24], which pointed out that populations of microorganisms were the largest in soils with the largest organic matter content and abundant nitrate, phosphate and potassium. The correlation between organic matter content and *A. flavus* propagules was also proven in other reports [25]. Gal Winter et al. [26] have provided a hypothesis that soils with high content of organic matter would have greater water retention capacity than those with a small content, thus providing a more suitable condition for fungal cultivation.

The pH measurement of the collected soil samples showed that the soil in Xiangyang peanut planting area was predominantly weakly acidic, which was mainly a naturally occurring geographical condition. However, the pH values of three samples from Yicheng were not the same, with pH of 4.95, 4.25 and 6.91, respectively. This situation may have been caused by the application of chemical fertilizers and other agricultural inputs during the operation. Additionally, the average number of *Aspergillus sp.* colonies in weakly acidic soil samples was much higher than that in weakly alkaline soil samples, so it could be inferred that *Aspergillus sp.* preferred to grow in an acidic environment. This result is consistent with that of a study conducted by Dadzie, M.A. et al., which provided a view that a low concentration of *A. flavus* in Akomadan may have been due to the observed relatively high alkaline soil pH [27]. From the perspective of soil pH, the average pH value of clay loam samples (5.33) was lower than that of sandy loam samples (6.13), which may have been responsible for the higher distribution of *Aspergillus* strains in clay loam. It has been reported that the most suitable pH value for aflatoxin synthesis is between 3.4–5.5 [28], and most of the soil in the peanut planting areas of Xiangyang is within this pH range, thus, more attention should be paid to the appropriate use of agricultural inputs in the process of peanut planting.

At present, there are few reports on the effects of soil types or texture on the distribution and toxin production capacity of *Aspergillus* strains. Some literature [12,29,30] proposes that sandy loam or light sandy soil is more conducive to the growth and reproduction of *A. flavus*. Wang [12], A.M. Torres et al. [30] and Windham G.L. et al. [31] believe that sandy loam is prone to high temperature and drought stress due to poor water retention, which increases the risk of peanuts or other crops being infected by *A. flavus* during the pod stage, while in clay soil with high water-holding capacity it is just the opposite. However, the survey conducted in Xiangyang showed that the density of *A. flavus* and the aflatoxin production capacity of strains in sandy loam were lower than that of those in clay loam, probably because the planting areas with sandy loam soil in Xiangyang are all along the Han River or the Tangbai River, where irrigation conditions are good, effectively alleviating the problem of poor water retention and drought stress. Moreover, good irrigation conditions also prevent the accumulated temperature of the soil from being too high so that *A. flavus* does not multiply in large numbers and peanuts are not susceptible to drought stress. On the contrary, the places where clay loam samples were collected were mostly hilly areas or hilly areas in northern Hubei, with sufficient sunshine and high accumulated temperature of the land, coupled with the good water retention of the clay loam, which created advantages for the propagation and toxin production of *A. flavus*.

## 4. Conclusions

Based on this survey in the peanut planting areas in Xiangyang, since the number of *Aspergillus sp.* colonies and the toxin production capacity of *Aspergillus* strains in clay loam were both higher than those in sandy loam, the peanuts growing in clay loam areas were at higher risk of contamination by *A. flavus* and aflatoxin, and more attention should be paid to the prevention and control of *A. flavus* in these areas during peanut growth. In order to reduce the risk of aflatoxin contamination of peanuts in Xiangyang, it is recommended that peanut varieties that are resistant to *A. flavus* be selected in high-risk planting areas in the future. Moreover, the application of a biocontrol fertilizer, strengthening field irrigation management and improvement of postharvest storage conditions should be considered when controlling peanuts’ contamination by *A. flavus* and aflatoxin.

## 5. Materials and Methods

### 5.1. Materials

#### 5.1.1. Reagents

The standards of aflatoxins, including AFB_1_, AFB_2_, AFG_1_ and AFG_2_, were purchased from Tanmo Quality Inspection Technology Co., Ltd. (Jiangsu, China). The methanol used in the high-performance liquid chromatography system was purchased from Thermalfisher Scientific (China) Co., Ltd. (Shanghai, China). The immunoaffinity column was purchased from Wuhan Huamei Biotech Co., Ltd. (Wuhan, China).

#### 5.1.2. Instruments

A Shimadzu LC-20AT high-performance liquid chromatography system (Kyoto, Japan) consisted of 4 solvent delivery units, one SIL-20A autosampler, one CTO-20A column oven, one RF-20A detector and one SPD-20A detector. The LC was in tandem with post-column photochemical derivatization, which was produced by Wuhan Trustworthy Technology CO., Ltd. (Wuhan, China).

#### 5.1.3. Samples

Soil samples were collected from 4 major peanut-producing areas in Xiangyang: Xiangzhou, Zaoyang, Yicheng and Gucheng. According to the actual distribution of peanut planting around the four main planting areas, we selected a total of 13 towns as sampling points, including two typical soil types which were clay loam from hilly land and sandy loam from the riparian zone. In the selected peanut planting area, soil samples were collected by using the five-point sampling method. Five subsamples were mixed into one sample and at least 3 samples were collected at each sampling point. The samples were collected by first brushing away the fallen leaves and deadwood on the top of land and taking the soil around the root interval about 10 cm deep. All the samples were marked according to the soil types after removing the large stones in the soil, and each sample was approximately 2 kg. At last, there were 35 soil samples collected in total, and the sampling distribution is shown in Figure 9. All the soil samples were collected during a peanut harvest period which is usually from the end of September to early October, in order to avoid differences of soil composition caused by the seasons.

### 5.2. Methods

#### 5.2.1. Isolation and Identification of *A. flavus*


Soil samples were dried, grinded and mixed thoroughly prior to use. Ten grams of soil was added to 90 mL of sterile water, mixed vigorously for 5 min in a constant-temperature shaker to make a sample base solution with a dilution of 10^−1^. Then, the base solution was serially diluted to 10^−2^. Twenty-five microliters of 10^−2^ dilution was spread on a dichloran-18% glycerol (DG-18) plate, and each sample was set to 2 replicates. The plates were incubated at 28 °C for 5 days. The yellow-green spore colonies grown on the DG-18 medium were counted and then picked and inoculated on *A. flavus* and *A. parasiticus* agar (AFPA) plates at 28 ± 1 °C for 3–5 days until individual colonies grew [20]. Isolates of *A. flavus* were preliminarily verified by bright orange coloration of the reverse colonies. Then, the isolates were sent to a third-party testing institution (Beijing Prime Sequencing Company (Wuhan, China)) for further molecular biological identification by sequencing with ITS universal primers. Universal primers ITS1(TCCGTAGGT-GAACCTGCGG) and ITS4(TCCTCCCGCTTATT-GATATGC) were used in PCR amplification.

Then, a small amount of mycelium was picked from *A. flavus* and *A. parasiticus* agar (AFPA) plates on DG-18 plates and incubated at 28 ± 1 °C for 5 days until yellow-green spores were obtained. The spores were washed with a 0.1% Tween 80 aqueous solution and stored in centrifuge tubes in a −40 °C freezer as a strain reserve.

#### 5.2.2. *Aflatoxin*-Producing Culture

Preserved *Aspergillus isolates* were inoculated in DG-18 plates for 5 days at 28 ± 1 °C. After incubation, the spores were washed with sterile water containing 0.1% Tween 80 to prepare a suspension of *Aspergillus sp.* conidia. The prepared conidia solution was placed under an electron microscope and counted with a hemocytometer plate. Then, a certain amount of spore suspension was aspirated and added to a conical flask containing 30 mL of liquid Sabouraud medium to a final concentration of 1.0 × 10^5^ spores/mL, and the flasks were incubated in a constant-temperature incubator shaker at 28 ± 1 °C for 7 days with 200 r/min rotating speed.

The toxin-producing culture solution was filtered with a sterilized gauze in a 15 mL centrifuge tube. One milliliter of toxin-producing culture solution was added into 4 mL of pure water and then passed through an immunoaffinity column, controlling the flow rate at 2–3 mL/min. The immunoaffinity column was first eluted with 10 mL of water 2 times and then eluted with 1 mL of methanol [11]. The methanol eluate was collected and detected by liquid chromatography (LC).

#### 5.2.3. Detection of Aflatoxin 

The method of detection used by Zhu et al. [9] was modified and used in detecting aflatoxins. The amount of aflatoxin in toxin-producing cultures was detected by high-performance liquid chromatography in tandem with post-column photochemical derivatization. The column was ZORBAX Eclipse XDB-C18 (5 μm-Micron, 4.6 × 150 mm) bought from Agilent Technologies (China) Co., Ltd., and the column temperature was 35 °C. The mobile phase was methanol and water (*V*:*V* = 45:55), and the flow rate was set as 0.9 mL/min. The detector was a fluorescence detector (excitation wavelength was 360 nm, emission wavelength was 440 nm). The injection volume was 10 μL.

#### 5.2.4. Calculation 

The calculation method reported in Zhang Xing’s survey [32] was modified and used.
(1)$$\mathrm{Number}\;\mathrm{of}\;\mathrm{colonies}\;\mathrm{of}\;Aspergillus\;flavus\;\mathrm{per}\;\mathrm{gram}\;\mathrm{of}\;\mathrm{soil}=\frac{\mathrm{number}\;\mathrm{of}\;\mathrm{colonies}\;\mathrm{on}\;\mathrm{the}\;\mathrm{plate}}{0.025\times \mathrm{dilution}\;\mathrm{factor}}$$
(2)aflatoxin production of *Aspergillus flavus* in one gram of soil (theoretical value) = number of *Aspergillus sp.* colonies per gram of soil × average amount of AFT produced by *Aspergillus* isolates

The average amount of AFT produced by *Aspergillus sp.* in Equation (2) is the amount of toxin in the culture solution measured after culturing the *Aspergillus* strains according to the method described in Section 5.2.2. In Equation (1), 0.025 is the volume of diluents added into DG-18 plates when culturing.

The comparisons of average numbers established in the text or tables were all analyzed by significance analysis, using Duncan’s new multiple range test to calculate the *p*-values.

## Figures and Tables

**Figure 1 toxins-14-00425-f001:**
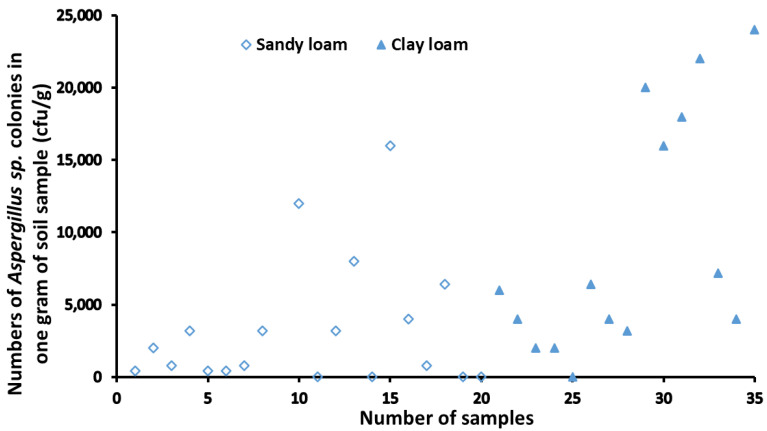
Numbers of *Aspergillus sp.* colonies in different soil types.

**Figure 2 toxins-14-00425-f002:**
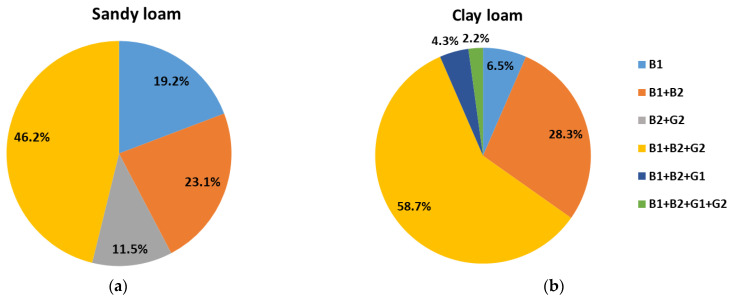
Distribution of toxigenic types of *Aspergillus isolates* from different soil types. (**a**) Toxigenic types of *Aspergillus isolates* from sandy loam; (**b**) toxigenic types of *Aspergillus isolates* from clay loam.

**Figure 3 toxins-14-00425-f003:**
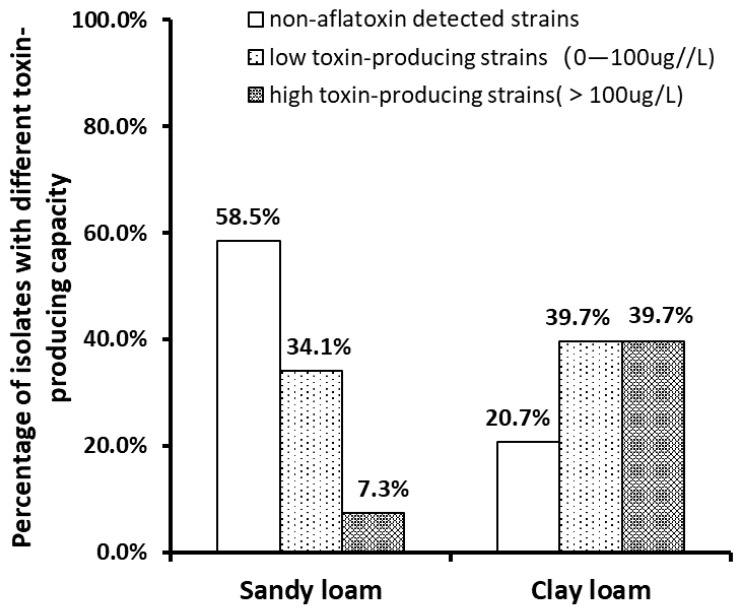
Distribution of *Aspergillus* isolates with different toxin production capacities in sandy loam and clay loam.

**Figure 4 toxins-14-00425-f004:**
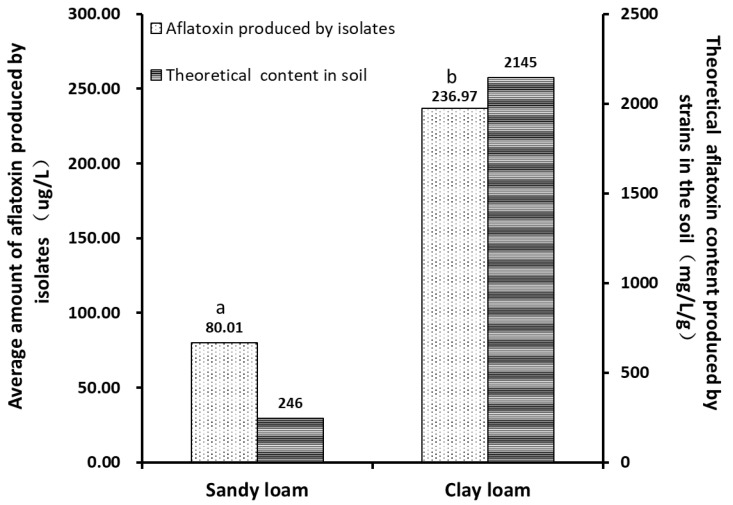
Toxin-producing capacity of *Aspergillus isolates* in sandy loam and clay loam. a, b indicate there is a significant difference between these two values.

**Figure 5 toxins-14-00425-f005:**
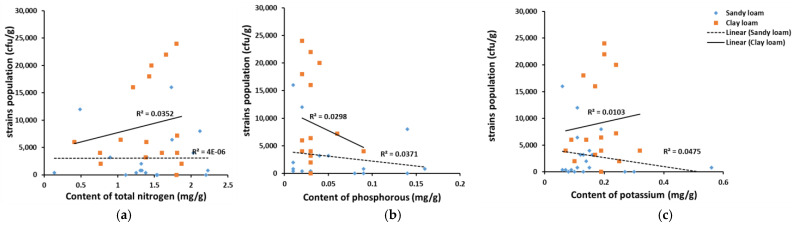
Correlation analysis of soil properties and *Aspergillus* strains population. (**a**) Correlation analysis of total nitrogen in soil samples and an *Aspergillus* strains population; (**b**) correlation analysis of phosphorous in soil samples and an *Aspergillus* strains population; (**c**) correlation analysis of potassium in soil samples and an *Aspergillus* strains population.

**Figure 6 toxins-14-00425-f006:**
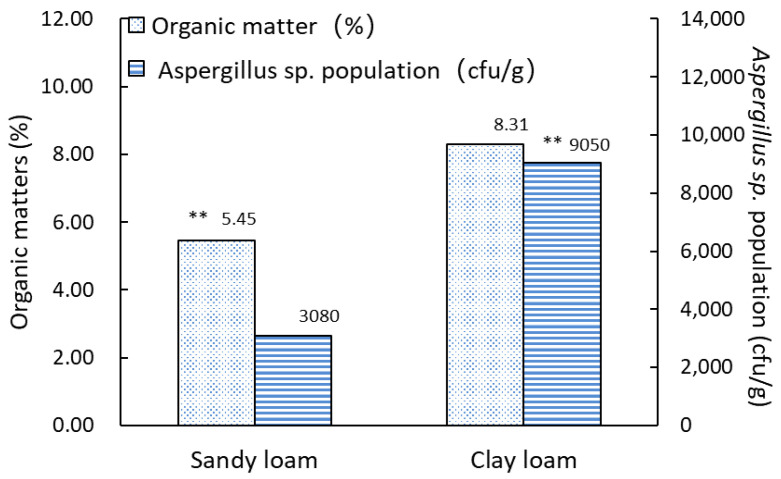
Organic matter and *Aspergillus sp.* populations in two soil types. ** represents a very significant difference.

**Figure 7 toxins-14-00425-f007:**
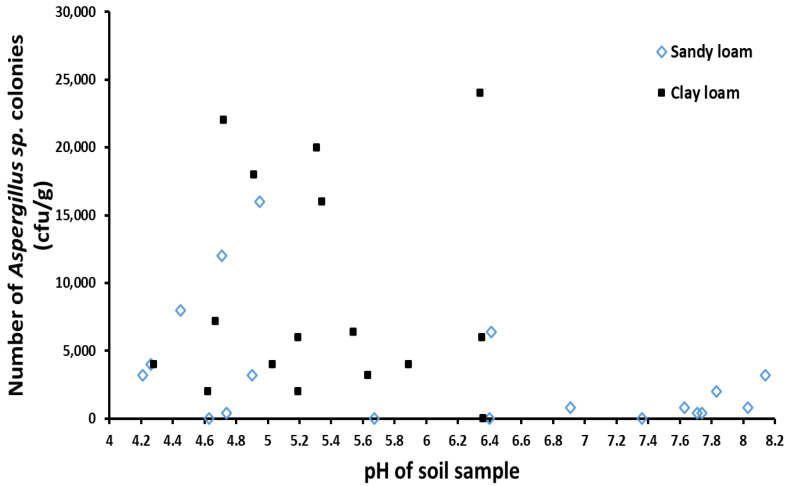
The relationship between the distribution of *Aspergillus sp.* colonies and soil pH.

**Figure 8 toxins-14-00425-f008:**
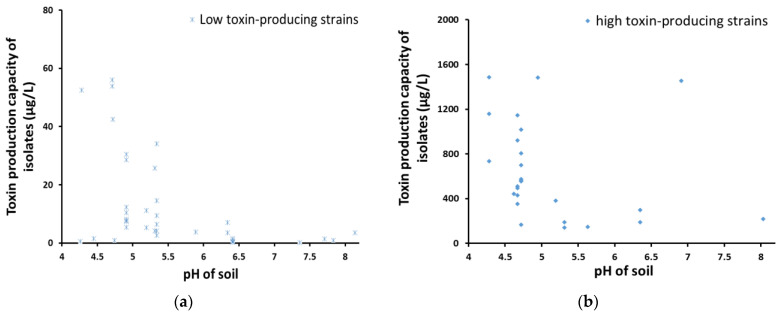
The relationship between the distribution of toxin-producing capacity of *Aspergillus* isolates and soil pH. (**a**) Toxin-producing capacity of low toxin-producing strains and soil pH; (**b**) toxin-producing capacity of high toxin-producing strains and soil pH.

**Figure 9 toxins-14-00425-f009:**
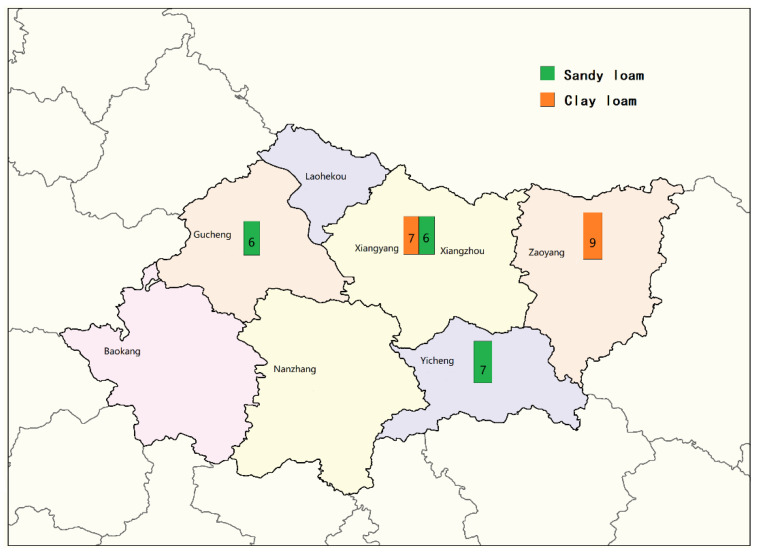
Sampling points’ distribution and number of samples in peanut planting areas in Xiangyang.

**Table 1 toxins-14-00425-t001:** Distribution of *Aspergillus sp.* colonies in different soil types.

Soil Type	Quantity of Soil Sample	Range of Colony Count (cfu/g)	Average of Colony Count (cfu/g)	Standard Deviation (SD)
Sandy loam	19	0–12,000	3080	4403
Clay loam	16	0–24,000	9050	8003
*p* value			0.0074 **	

The *p*-value was calculated by Duncan’s new multiple range test. **, indicates a very significant difference.

**Table 2 toxins-14-00425-t002:** Distribution of *Aspergillus* isolates in sandy loam and clay loam and the amount of toxin production.

Soil Types	Numbers of Strains		The Average Amount of Aflatoxin Produced by the Strains (μg/L)		The Range of the Amount of Aflatoxin Produced by the Strains (μg/L)	The Theoretical Amount of Aflatoxin Produced in the Soil (mg/L/g)
	Non-Aflatoxin Detected Strains	Toxin-Producing Strains	Average	SD		
Sandy loam	24	17	80.01	320.53	0–1482.81	246.44
Clay loam	12	46	236.97	336.09	0–1485.16	2144.58
*p* values			0.029 *			

* indicates a significant difference.

**Table 3 toxins-14-00425-t003:** Major properties of different types of soil and the number of *Aspergillus sp.* colonies.

Soil Type	Organic Matter (%)		Total Nitrogen (mg/g)		Phosphorous (mg/g)		Potassium (mg/g)		pH		Average of Colony Count (cfu/g)	
	Average	SD	Average	SD	Average	SD	Average	SD	Average	SD	Average	SD
Sandy loam	5.45	1.71	1.45	0.53	0.05	0.05	0.15	0.11	6.07	1.48	3080	4403
Clay loam	8.31	0.86	1.39	0.44	0.03	0.02	0.18	0.07	5.34	0.65	9050	8003
*p* value	0.0001 **		0.7726		0.1331		0.4205		0.0784		0.0074 **	

*p* value was calculated by Duncan’s new multiple range test. ** represents a very significant difference.

**Table 4 toxins-14-00425-t004:** The relationship between *Aspergillus sp.* colonies’ distribution and toxin-producing capacity and soil pH.

pH of Soils	Numbers of Strains		Number of Colonies		The Average Amount of Aflatoxin Produced by the Strains (μg/L)		The Range of the Amount of Aflatoxin Produced by the Strains (μg/L)
	Non-Aflatoxin Detected Strains	Toxin-Producing Strains	Average	SD	Average	SD	
<7 (weakly acidic)	34	58	7100	6885	182.6	363.5	0–1485.16
>7 (weakly alkaline)	2	5	1086	1124	32.3	82.7	0–219.8
*p* value			0.015 *				

* represents a significant difference.

## Data Availability

Data available on request from the authors.

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
