# Peer review of "Discovery of the Relationship between Distribution and Aflatoxin Production Capacity of Aspergillusspecies and Soil Types in Peanut Planting Areas"

_toxins, 2022, doi:10.3390/toxins14070425_

Round 1

Reviewer 1 Report

The topic is very interesting and within the scope of this Journal.

In any case, I think it has serious deficiencies in the methodology. The method for distinguishing toxigenic and non-toxigenic strains is not specified. It is not enough to analyze the concentrations of aflatoxins, molecular biology studies will have to be carried out. Sometimes the results are not statistically confirmed.

Keywords: “Production capacity” should be removed because more information is not provided.

Introduction

Line 62 and 63: I don´t undertand well the scope of the work because you write: “Xiangyang were studied, in order to discover the specificity of soil and its connection with the distribution  and aflatoxin production capacity of Aspergillus flavus, and to provide more solutions for future development of prevention and control (and you don´t study in this resarch prevention and control measures and their impact on aflatoxins) of aflatoxin contamination in peanuts” Ok, but How do you select the solutions? Did you also find something related to this affirmation in the scientific literature? It sholud be necessary to write a paragraph directed to these measures. It is important to justify this study if there is a way to prevent it, it is important to have alternatives to avoid mycotoxin contamination which have useful this study.

Results and discussion/Conclusions:

Line 87: “was significantly lower than that in clay loam”: This affirmation has to be justified by statistical results.

Line 104: The title is not correct: “Afaltoxin”

Lines 178,180, 187: All these figures are not confimed statistically.

Line 191: There aren´t studies out of China. It would be enriching to include them.

Line 284:  What do you mean with good irrigation?

Line 297: You haven´t investigated  resistance genes and biocontrol. I don´t understand these conclusions.

Materials and methods

I don´t know how is the procedure to distinguish toxigenic and non toxigenic strains. Please, specify it. I think it is a major deficiency in the manuscript.

Line 306: TMRM should be written in detail.

Line 307: “Thermalfisher Scientific (China) Co., Ltd.” It is confusing, please put this reference in a correct way. The city of China is not included.

Line 325: The weight of the samples is not considered, please, specify.

Line 351: “200 r/min” something is missing.

Line 352: “15ml” it should be changed to “15 mL”.

Line 357: The references related to chromatographic equipment are not included.  Please, complete it.

Line 360: “Photochemical derivatization” is not explained in detail, please, complete it.

Line 365 and 366. I don´t undertand the calculations in this way. Please, try to explain (1) and (2). Try to explain it easier.

Line 367: I can´t see equation 2.

Line 367: A. flavus should be written in italics

Line 368: A. flavus should be written in italics

References:

Please, have a look at the references. It should be improved. There are some deficiencies. It is not well reviewed.

Lines 390, 392 : Some surnames are in capital letters and should be changed.

Lines 403 and 428: “et al” is not correct, please complete it

Line 412: Maybe the surnames are wrong

Reviewer 2 Report

The manuscript investigated the relationship between distribution and aflatoxin production capacity of Aspergillus flavus and soil types. The results are relevant for the mycotoxin community and important to understand the accumulation of mycotoxins also in asymptomatic plants due to the soil uptake.

Results:

-Table 1, 2, 3 4 I would suggest also adding the SD when showing the average value to understand the data distribution

-Figure 4, 5 and 7 I would suggest to add the statistical analysis to clearly see if any comparison is statistically significant.

-Line 66-68: please delete this part from the author’s guidelines “This section may be divided by subheadings. It should provide a concise and precise 66 description of the experimental results, their interpretation, as well as the experimental 67 conclusions that can be drawn.”

Methods:

-Line 358: I would suggest improving the “Detection of aflatoxin” paragraph. What the author mean with “liquid chromatography tandem”. Which column was used? It is just stated “The column was C18(5μm,4.6mm*150mm)”.

-Statistical analysis is not described, please add this paragraph.

Line 321-322: Five sub-samples were mixed into one sample and at least 3 samples were collected at each sampling point. When was performed the sampling? In which season? Would this influence the soil composition? I would suggest to add these details and comment on the possible differences occurring when different sampling time point will be considered.

Reviewer 3 Report

This MS can be very good and useful for science and farming as well.

The design of the experiments and the scientific methods used are appropriate.

The reviewer understands that this work is basically looking for a relationship between the presence of Aspergillus species and their toxin production in terms of soil properties, but the idea that these fungi are not only soil fungi but also endophytic fungi would be worth a sentence or two. The pH and humus content of the soil can have a significant effect on the phenological and physiological properties of the host plant and, indirectly, on the endophytic fungi living in it.

On the other hand, weakly acidic soils are generally more favorable for soil fungi, which use exoenzymes for external digestion, so they are chylotrophic organisms.

The results are clear, their description is professional and understandable.

However, there are a number of minor errors, omissions, or inaccuracies in the manuscript that are indicated and suggested:

Ad 23: the species under study is recommended to be given here, also marked with an author at the beginning of the text, with a full scientific name.

Missing links:

Ad 380 .:… Scientific reports, 6 (1), 1-12.

Ad .: 384: International Journal of Environmental Research and Public Health, 17 (21), 7850.

Ad 388-389 Is this a book? Place of publication?

Ad 393 "aflatoxin production"

Ad 399 Is this a book? Place of publication?

Ad 400:… Journal of Fungi, 7 (5), 381.

 Ad 411: Is this a book? Place of publication?

Ad 412 Proctor, R. H.… ..

Latin names not spelled according to botanical nomenclature (no emphasis):

Ad 69, 77, 104, 138, 187 (Fig. 9a. X-axis), 368

Ad 388, 393, 395, 397, 401, 407, 416, 418, 422, 426, 430

Letter, character and separation errors:

Ad 33:…, etc [7]….

Ad 49… crops [12]….

Ad 51… crops [13-18], and…

Ad 52… soil types [8, 19]….

Ad 88: Figs. 2. In the inscription on the X-axis: "… conolies"

Ad 104: Afaltoxin

Ad 123 "(236.7μg / L)"

Ad 125 "(80.01μg / L)"

Ad 131 / Fig 4. At the top of the columns [ug // L) + the numbers are partially covered.

ad 171 "… samples (pH = 6.14)…"

ad 196 "ing Xing [11] 's…"

ad 242: "… production [20, 21]…."

Ad 250 "[al [22], w…"

Ad 253 „… reports [23]. Gal Winter et al [24] have… ”

Ad265 „… by Dadzie M A et al, w…”

Ad 276 „… Wang [12], A.M. Torres et al. [28] and Windham G L et al. [29] b… ”

Ad 361 ”C s C18 (5μm , 4.6mm * 150mm),…”

Ad 362 "… water (V : V = 45:55), and…."

Ad363 "… 0.9mL / min”. "

Ad 364 "… was 440nm) and the injection volume was 10μL.”. "

Ad 366-367 „… theoretical value) =” (partly hidden)

Ad 369 „… method described in 1.2. ... ... ”

Ad 391 "aflatioxins".

Ad 394: "41 (02): p"

Ad 396: “. 2013: China. ” ?

Unclear term, wording, reference:

Ad 48… the root soil…

Ad 53… between / among? ...

Ad 66-68 Deletion is recommended.

Ad 69:… described in 1.2.1, 116 strains….

Ad 73: Proposed: biological identification.

Ad 76: It is proposed to clarify: "… in the database."

Ad 105 ”… method described in 2.2.2,…”

Author Response

This manuscript is a resubmission of an earlier submission. The following is a list of the peer review reports and author responses from that submission.